# Phosphoinositide Signaling in Immune Cell Migration

**DOI:** 10.3390/biom13121705

**Published:** 2023-11-24

**Authors:** Ruchi Kakar, Chinmoy Ghosh, Yue Sun

**Affiliations:** Department of Oral and Craniofacial Molecular Biology, Philips Institute for Oral Health Research, School of Dentistry, Virginia Commonwealth University, Richmond, VA 23298, USA; kakarr@vcu.edu (R.K.); ghoshc@vcu.edu (C.G.)

**Keywords:** phosphoinositides, phosphoinositide kinase, phosphatase, immune cells, migration, polarization, neutrophil, macrophage, B cell, T cell

## Abstract

In response to different immune challenges, immune cells migrate to specific sites in the body, where they perform their functions such as defense against infection, inflammation regulation, antigen recognition, and immune surveillance. Therefore, the migration ability is a fundamental aspect of immune cell function. Phosphoinositide signaling plays critical roles in modulating immune cell migration by controlling cell polarization, cytoskeletal rearrangement, protrusion formation, and uropod contraction. Upon chemoattractant stimulation, specific phosphoinositide kinases and phosphatases control the local phosphoinositide levels to establish polarized phosphoinositide distribution, which recruits phosphoinositide effectors to distinct subcellular locations to facilitate cell migration. In this Special Issue of “Molecular Mechanisms Underlying Cell Adhesion and Migration”, we discuss the significance of phosphoinositide production and conversion by phosphoinositide kinases and phosphatases in the migration of different types of immune cells.

## 1. Introduction

Cell migration is a physically integrated molecular process in which cells move from one location to another [1]. This process plays pivotal roles in multicellular organisms for various physiological and pathological events, including embryonic development, immune responses, tissue repair, and cancer metastasis. In the context of the immune system, the function of most immune cells relies on their migratory ability to detect and combat infections, clear foreign substances, and maintain tissue homeostasis. To initiate an immune response, multiple types of immune cells need to traffic between different tissues and lymphoid organs via the circulatory and lymphatic systems. Neutrophils and monocytes circulate in the blood and can directly migrate to sites of inflammation or tissue damage to rapidly carry out their immune functions. Dendritic cells (DCs) are present in various tissues throughout the body, where they capture antigens from pathogens, damaged cells, or other sources. After capturing antigens, DCs undergo a maturation process and then migrate from the peripheral tissues to enter regional lymph nodes, where the DCs present the captured antigens to T cells and B cells. Naïve T cells always first encounter their cognate antigen in secondary lymphoid organs such as lymph nodes, which activate naïve T cells and induce their full differentiation. The differentiated effector T cells are released from the lymph nodes, which then migrate to the peripheral sites of infection or inflammation. In summary, cell migration allows immune cells to reach infection sites, interact with other immune cells, and carry out their specific functions.

There are two distinct modes of cell migration, mesenchymal and amoeboid [2,3,4]. For mesenchymal migration, cells form focal adhesions with the extracellular matrix (ECM) and maintain an elongated morphology with distinct leading and trailing edges. The epithelial-derived and solid tumor cells normally undergo mesenchymal migration for their movement. For amoeboid migration, cells do not form extensive focal adhesions with the ECM, and the migrating cells show a rounded or irregular cell morphology with a highly contractile rear part, known as the uropod [5]. Immune cells mostly adopt amoeboid migration [3]. During migration, immune cells undergo cycles of expansion and contraction, which allow them to squeeze through gaps in the ECM.

Phosphoinositides (PI) are a family of phospholipid molecules that play crucial roles in various cellular processes, including signal transduction, vesicular trafficking, cytoskeleton rearrangement, and cell polarization. The phosphoinositide signaling system is composed of a network of phosphoinositides, phosphoinositide kinases, phosphatases, and phosphoinositide-binding effectors [6]. There are seven species of phosphoinositides distributed to distinct subcellular locations, where they recruit their specific effectors to mediate broad biological effects [6] (Figure 1A,B). These phosphoinositides can interconvert to each other using their specific kinases and phosphatases [7]. Phosphoinositides play significant roles in immune cell signaling and immune responses [8,9,10]. In the context of immune cell migration, phosphoinositides can directly modulate immune cell migration by controlling cell polarization, actin retrograde flow, and uropod retraction. Phosphoinositides can also indirectly regulate immune cell migration by controlling immune cell activation and chemoattractant cytokine production. In this review, we will focus on the current knowledge of the direct function of phosphoinositides in modulating immune cell migration. Additionally, phosphoinositides can also be hydrolyzed by different kinds of phospholipases, producing secondary messengers to regulate immune cell responses [11]. The significance of phospholipases and the production of secondary messengers in the immune system has been well defined in other reviews [11,12]. For the present review, we will be limiting our discussion to the production and conversion of different phosphoinositides by specific kinases and phosphatases during immune cell migration. The function of phospholipases and secondary messengers will not be further discussed in this review.

## 2. Role of Phosphoinositide 3-Kinases (PI3Ks) in Immune Cell Migration

PI3Ks are a family of kinases that phosphorylate the 3-position of the inositol ring of phosphoinositide lipids. Based on their structure and substrate specificity, PI3Ks can be grouped into three major classes: class I, class II, and class III PI3Ks [13] (Figure 2A). Class I PI3Ks are composed of a regulatory subunit and a catalytic subunit. Mammals express four class I catalytic isoforms named p110α, β, γ, and δ, which use PI4,5P_2_ as a substrate to generate PI3,4,5P_3_. The p110α and p110β proteins are ubiquitously expressed in all somatic cells, whereas p110γ and p110δ are predominantly expressed in leukocytes and play multiple roles in immune responses [14]. Unlike class I PI3Ks, class II PI3Ks (isoforms α, β, γ) have only catalytic subunits, which produce two different phosphoinositides, PI3P and PI3,4P_2_ [15]. VPS34 is the sole member of class III PI3K that phosphorylates phosphatidylinositol to synthesize PI3P [16,17]. Extensive studies have been carried out to understand the role of class I PI3Ks in immune cell migration [18,19,20]. However, to date, the function of class II or class III PI3Ks in immune cell migration has not been well characterized.

A chemotactic stimulus can activate class I PI3Ks at the leading edge, where class I PI3Ks produce PI3,4,5P_3_ to recruit and activate PI3,4,5P_3_ effectors. These PI3,4,5P_3_ effectors control cell migration by modulating cell polarization, adhesion, protrusion formation, and cytoskeletal rearrangement. The following sections discuss the importance of PI3Ks in immune cell migration in detail.

### 2.1. PI3K and Neutrophil Migration

Neutrophils are the most abundant leukocytes and form the first line of defense against pathogens. Migrating to the site of infection is extremely crucial for neutrophils and other immune cells to mount an immune response. One of the inherent characteristics of neutrophils is chemotaxis, a type of directional migration, where cells sense a chemoattractant, attain polarization, and migrate toward the chemoattractant. Neutrophils are highly motile and have less stable adhesive contact with the substratum and lack stable actin cables compared with adhesive cells, thus endowing them with high migratory speed [21].

Neutrophils can display strong chemotactic responses even when the gradients are extremely low [22]. Class I PI3K plays critical roles in chemoattractant gradient sensing and cell polarization. Upon chemoattractant stimulation, class I p110δ and p110γ (primarily activated by G protein-coupled receptors (GPCRs)) are activated to mediate PI3,4,5P_3_ production at the leading edge [21,23,24], whereas the phosphatase and tensin homolog (PTEN) is recruited at the rear to induce the hydrolysis of PI3,4,5P_3_ [25] (Figure 2B). The polarized PI3,4,5P_3_ distribution facilitates the recruitment of PI3,4,5P_3_ effectors such as Akt/PKB, Rac, and Cdc42 to the leading edge [21,26,27,28,29]. These effectors control F-actin polymerization at the leading edge to maintain cell polarity and provide the force for membrane protrusion.

Both class I PI3Kγ and PI3Kδ are required for N-formyl-Met-Leu-Phe (fMLP)-induced neutrophil chemotaxis. PI3Kγ plays an important role in the preliminary stage during the initial burst of fMLP-induced PI3,4,5P_3_ biosynthesis and later on, PI3Kδ is involved in the amplification of PI3,4,5P_3_ that leads to polarization and chemotaxis [22]. PI3Kγ^−/−^ neutrophils were unable to migrate across a chemotactic gradient of fMLP. These neutrophils demonstrated reduced cell motility along with a loss of directionality [30,31]. Loss of PI3Kγ led to a lack of F-actin–Akt colocalization at the leading front compared with wild-type neutrophils. The lack of localized fMLP-induced PI3,4,5P_3_ in these PI3Kγ^−/−^ neutrophils could be the reason for the reduced sensing to the chemotactic gradient and the inability to stabilize a leading front [30]. Blocking PI3Kδ function by a selective inhibitor IC87114 led to a 60–65% reduction in fMLP-stimulated PI3,4,5P_3_ production, which caused inhibition of polarized morphology and decreased CD18 integrin-dependent chemotactic migration in neutrophils [22]. However, random neutrophil movement was not impacted by inhibiting PI3Kδ, neither was F-actin synthesis blocked or adhesion affected. These observations highlighted that PI3Kδ had a limited role in neutrophil polarization and directional migration and did not regulate all aspects of neutrophil movement.

There are some contradictory reports of the effects of PI3K inhibitors or loss of PI3K on neutrophil migration. A study reported that PI3K is important only for the timely migration of neutrophils toward fMLP [32]. Long-term migration studies showed that PI3K-deficient cells also demonstrated chemotaxis toward fMLP; however, the process was delayed. Cells treated with a pan-PI3K inhibitor also demonstrated delayed chemotaxis compared with untreated cells. The authors cite that this characteristic could be attributed to a delay in integrin regulation as neutrophils treated with a PI3K inhibitor or those that are PI3Kγ deficient failed to show upregulated adhesion to ICAM-1 (an integrin ligand) [32]. Another study reports that PI3Kγ-dependent accumulation of PI3,4,5P_3_ at the leading edge is not required for efficient gradient-sensing and gradient-biased neutrophil movement [33]. Contrarily, PI3Kγ activity is required for fMLP-stimulated random migration in neutrophils [33]. One possible explanation for these contradictory results is that the PI3K function in neutrophil migration is context-dependent. The different states of priming of the neutrophils or different types of surfaces on which neutrophils migrate can affect the dependence on PI3K [33]. For example, PI3K inhibitors can block fMLP-induced migration in unprimed neutrophils; whereas, PI3K inhibitors have no effect on fMLP-induced migration in the LPS-primed neutrophils [33].

In vivo studies support that PI3K can control neutrophil migration in response to some pathological conditions. Neutrophil transepithelial migration into the alveolar space was reduced in PI3Kγ^−/−^ mice in response to LPS-induced lung injury [34]. Additionally, treatment with the PI3Kγ inhibitor AS-605240 also reduced the LPS-induced neutrophil migration in vivo, which further confirms the important roles of PI3Kγ in neutrophil migration [34]. Knockdown studies in zebrafish demonstrated that PI3K knockdown reduced the migration of neutrophils towards wounds [23]. Similar results were seen when PI3Kγ-specific inhibitor AS-605240 was used, confirming that PI3Kγ was vital in neutrophil recruitment to the wounds. PI3K inhibition led to a loss of stable F-actin polarity, which negatively affected neutrophil migration to the wounds [23].

Rac belongs to the Rho family of GTPases and plays a critical role in generating actin-rich lamellipodial protrusion to provide the driving force for migration [35,36]. There are three Rac isoforms: Rac1, Rac2, and Rac3. Both Rac1 and Rac2 are reported to be required for neutrophil chemotaxis [37,38]. DOCK2 is a Rac guanine exchange factor (GEF) that is required for Rac activation in neutrophils [39]. DOCK2 is also a PI3,4,5P_3_ effector that associates directly with PI3,4,5P_3_ and can translocate to the leading edge in response to chemoattractant stimulation [39]. Knockout of PI3Kγ impaired chemoattractant-induced DOCK2 recruitment to the leading edge of neutrophils, which confirmed the requirement of PI3Kγ and PI3,4,5P_3_ production for DOCK2 translocation [40]. Neutrophils from DOCK2^−/−^ mice showed impaired activation of Rac1 and Rac2, which led to the blocking of polarized F-actin accumulation at the leading edge. Therefore, it can be seen that the loss of the PI3,4,5P_3_ effector DOCK2 inhibited neutrophil chemotaxis [39].

An interesting study compared directional migration in neutrophils isolated from old and young healthy donors [41]. In younger donors, the PI3K activation level in resting neutrophils was low and chemoattractants could rapidly induce PI3K activation, following which the activation levels dropped quickly. Whereas, neutrophils from older donors had constitutively high active PI3K levels throughout and the change in PI3K activation status was insensitive to chemoattractant stimulation. The neutrophils from older donors showed decreased directional migration (chemotaxis) without affecting random migration (chemokinesis) in response to chemoattractant gradients when compared to neutrophils from younger donors. The authors hypothesize that blocking PI3Kγ or PI3Kδ constitutive activation with PI3K inhibitors could likely improve directional migration in neutrophils from old donors [41]. These results suggest that both timely activation and inactivation of PI3K are required for normal neutrophil directional migration. However, the mechanism leading to changes in PI3K activation during aging remains unclear.

### 2.2. PI3K and Macrophage Migration

Macrophages are derived from monocytes and play an important role in maintaining tissue homeostasis, antigen presentation, and cytokine production [42]. Macrophages are professionally motile cells and they must be able to migrate to specific sites upon demand to perform their function in response to infection and inflammation, wound healing, and surveillance [43,44]. PI3Kγ controls migration in macrophages. Macrophages from PI3Kγ^−/−^ mice showed reduced migratory speed and loss in their chemotactic ability in response to different chemoattractants such as MCP-1, RANTES (regulated on activation, normal T cell expressed and secreted), macrophage inflammatory protein-5 (MIP-5), macrophage-derived chemokine (MDC), stromal cell-derived factor-1 (SDF-1), C5a, and VEGF [45]. In vivo animal model studies suggest that PI3Kγ can affect the immune disease development by modulating macrophages and neutrophil migration. Rheumatoid arthritis (RA) is an autoimmune disease characterized by chronic inflammation of peripheral joints, articular cartilage and bone, and infiltration of leukocytes into the inflamed tissue [46]. Knockout of PI3Kγ in mice reduced macrophage migration into bone joints, thus alleviating the symptoms of rheumatoid arthritis [47]. The decreased migration was related to the inability of PI3Kγ^−/−^ macrophages to induce phosphorylation of Akt on chemoattractant stimulation. Similarly, treating mice with the PI3Kγ-specific inhibitor AS-605240 led to reduced joint swelling compared with mice injected with saline alone [47]. This suggests that PI3Kγ can be used as a target to treat immune diseases.

The class I PI3K regulatory subunit p85α was also reported to play an important role in macrophage migration. Bone marrow-derived macrophages that were deficient in p85α demonstrated a significant reduction in wound closure compared with their wild-type counterparts [48]. Additionally, reduced migration to M-CSF was seen and was found to be associated with reduced adhesion and migration on ECM proteins like fibronectin and vascular cell adhesion molecule-1 [48]. The p85/p110α isoform of PI3K is needed for chemoattractant-induced actin cytoskeletal rearrangement [48]. Cytoskeletal proteins such as Cas, FAK, ezrin, and actin-binding protein profilin are known to interact with p85, which could be the reason why p85^−/−^ mice have cytoskeletal defects [48].

### 2.3. PI3K and B-Cell Migration

B cells play a central role in the adaptive immune response by producing antibodies that can recognize and neutralize pathogens such as bacteria, viruses, and other foreign invaders [49,50]. Treatment with the PI3K inhibitor wortmannin significantly reduced macrophage inflammatory protein-1α (MIP-1α)-induced B-cell migration, which indicated that PI3K was indispensable for MIP-1α-stimulated B-cell migration [51]. A recent study demonstrated how inhibition of different isoforms of PI3K affected B-cell migration. Knockdown of PI3Kγ or treatment with the PI3Kγ-specific inhibitor CZC24832 reduced the migration of B cells [52], whereas, inhibiting both PI3Kγ and PI3Kδ isoforms using duvelisib had more remarkable effects than using a single-isoform selective inhibitor in JVM3 cells, a B-cell cell line [52], which suggests that both PI3K isoforms have an important function in B-cell migration. In mouse B cells, the chemotactic response to the chemokine CXCL13 was reduced when animals were deficient in p110δ or expressed mutant p110δ [53,54]. This further supports the important roles of PI3K in B-cell migration.

### 2.4. PI3K and T-Cell Migration

The function of PI3K in T-cell migration is context-dependent. Intracellular PI3,4,5P_3_ levels in T cells are low but increase considerably following T-cell receptor triggering by antigenic peptides [55]. Recruitment of T cells in lymph nodes requires chemokines such as CCL19 and CCL21, ligands of the CC chemokine receptor CCR7 [56]. In the lymph nodes, unstimulated T cells showed homogenous basal PI3K activity through the cytosol and nuclei, whereas on using anti-CD3 and anti-CD28 coated beads, T cells showed increased PI3K activity at the cell periphery. Conversely, when migrating toward CCL19, the Akt enhancement in the majority of T cells was similar to that seen by unstimulated cells, which did not follow the conventional Akt increase seen during migration in other cell types [56]. The lack of a 3′-phosphoinositide amplification loop downstream of chemokine receptors may prevent PI3K activation after chemokine signaling. This suggests that PI3K-independent routes might be present in T cells in the lymph node environment to mediate T-cell migration. It was also reported that the chemokine receptor CXCR3- and CCR4-mediated T-cell migration was not affected by PI3K inhibitors [57,58]. PI3Kδ is not required for constitutive or chemokine-induced T-cell migration [59], whereas, T-cell receptor (TCR) can mediate PI3Kδ activation, which is required for antigen-induced T-cell migration and tissue infiltration [59]. Treatment with PI3K inhibitors blocked antigen-dependent T-cell migration and T-cell trafficking to antigenic sites [59].

## 3. PIPKIs and PI4,5P_2_ in Immune Cell Migration

PI4,5P_2_ is involved in PI3K-mediated signaling by serving as a substrate of PI3K to produce PI3,4,5P_3_ [60]. It can also be hydrolyzed by phospholipase C to generate the secondary messengers diacylglycerol and inositol 1,4,5 trisphosphate [61]. PI4,5P_2_ can directly regulate a wide range of cellular functions by interacting with and modulating its effector proteins [62]. The majority of cellular PI4,5P_2_ is produced by type I phosphatidylinositol-phosphate kinases (PIPKIs) that preferentially utilize PI4P as the substrate to generate PI4,5P_2_ [63]. The PIPKI families are composed of three isoforms α, β, and γ, and are encoded by different genes [63]. The PIPKI genes undergo alternative splicing to generate multiple splice variants, which confer diversity to PIPKIs [62,64]. Different PIPKIs can interact with their specific effector proteins, which recruit them to distinct subcellular locations to perform specific biological functions.

It has been well established that PI4,5P_2_ and PIPKIs can modulate mesenchymal migration by regulating local PI3,4,5P_3_ levels and by directly controlling PI4,5P_2_-effector function. During mesenchymal migration, PI4,5P_2_ and PIPKIs regulate adhesion turnover, actin cytoskeleton dynamics, and cell polarization. Although the function of PI4,5P_2_ and PIPKIs in immune cell migration has received less attention, there are some reports that support that these PIs could also play an important role in immune cell migration.

PIPKIγi2, an alternative splice variant of PIPKIγ, binds to the PI4,5P_2_-effector protein talin. By producing PI4,5P_2_, PIPKIγi2 promotes β-integrin–talin interactions to stimulate integrin activation. During EGF-induced mesenchymal migration, PIPKIγi2 was recruited to the leading edge, where it was required for the nascent adhesion formation and for protrusion extension. In contrast, during neutrophil migration, PIPKIγi2 and PI4,5P_2_ generation were seen to be enriched in the uropods at the rear upon chemoattractant stimulation and was persistent throughout chemotaxis [65] (Figure 3). Expression of kinase-dead PIPKIγi2 inhibited uropod formation and prevented rear retraction, suggesting a role of PI4,5P_2_ in the neutrophil “backness” response during chemotaxis [65]. Another research reported that depletion of PIPKIγ blocked the production of PI4,5P_2_ at uropods and impaired neutrophil recruitment in vivo [66]. PIPKIγ was required for polarized RhoA GTPase activation and chemoattractant-induced integrin activation. Therefore, the loss of PIPKIγ impaired neutrophil adherence to endothelial cells [66].

Another PIPKI isoform, PIPKIβ, was also reported to localize to the uropod in the polarized human neutrophil cell line HL60, and this localization required the C-terminus of PIPKIβ [67]. The loss of PIPKIβ impaired RhoA activation, neutrophil polarity, and chemotaxis. The C-terminal tail of PIPKIβ could interact with EBP50 (4.1-ezrin-radixin-moesin (ERM)-binding phosphoprotein 50) to further enable interactions with ERM proteins and Rho-GDP dissociation inhibitor (RhoGDI) [67] (Figure 3). These processes triggered RhoA activation and were required for uropod function and neutrophil polarity and chemotaxis.

## 4. Other Phosphoinositide Signaling in Immune Cell Migration

Apart from PI3,4,5P_3_ and PI4,5P_2_, other phosphoinositides such as PI3,4P_2_, PI4P, and PI3,5P_2_ are also reported to play different functions in immune cell migration. During neutrophil migration, PI3,4P_2_ is distributed in a back-to-front gradient, which is required for neutrophil polarization [68]. Macropinosomes containing PI3,4P_2_ are formed in the front and then transported along the microtubules to the rear [69] (Figure 4). The macropinosomes finally break apart and fuse with the plasma membrane at the rear [69]. This helps the neutrophils maintain the PI3,4P_2_ back-to-front gradient during migration.

PI3,4P_2_ can be produced by class I PI3K or class II PI3K using PI4P as a substrate. However, it remains unclear which type of PI3K is critical in maintaining the PI3,4P_2_ back-to-front gradient. PI3,4P_2_ can also be generated via phosphoinositide 5-phosphatases by the dephosphorylation PI3,4,5P_3_ [70]. SHIP1 and SHIP2 belong to the SHIP (Src homology 2 (SH2) domain-containing inositol-5-phosphatase) family of phosphatases that can convert PI3,4,5P_3_ to PI3,4P_2_. SHIP1 expression is restricted to leukocytes and spermatocytes, whereas SHIP2 is ubiquitously expressed [71,72]. Both SHIP1 and SHIP2 control neutrophil migration. Genetic depletion of SHIP1 causes severe defects in neutrophil polarization and chemotaxis [73]. Loss of SHIP1 led to the diffused distribution of PI3,4,5P_3_, which impaired the PI3,4,5P_3_ gradient formation and neutrophil polarization in response to chemoattractant stimulation [73]. In addition, SHIP1-knockout neutrophils were extremely adherent, which decreased their chemotactic ability [74]. SHIP2 genetic inactivation significantly decreased PI3,4P_2_ levels in neutrophils [75]. SHIP2-defective neutrophils also showed significant chemotactic defects and their recruitment to the sites of inflammation was also blocked [75].

PI3,4P_2_ can also regulate B-cell migration. PI3,4P_2_ was present in high amounts in the plasma membrane of migratory B cells with locally enhanced levels at the front [76]. Lamellipodin is a PI3,4P_2_ effector that binds to PI3,4P_2_ with its PH domain [77]. As an important cytoskeletal regulator, Lamellipodin supports lamellipodia protrusion and cell migration [78]. Lamellipodin colocalized with PI3,4P_2_ on the plasma membrane of B cells and the knockdown of Lamellipodin significantly impacted directional migration, which indicated the role of Lamellipodin and PI3,4P_2_ in B-cell migration [76]. Expression of PI3,4P_2_-specific phosphatase inositol polyphosphate 4-phosphatase led to the depletion of PI3,4P_2_ in B cells, which reduced the overall motility and migration directionality in response to a chemoattractant [76]. Cells with PI3,4P_2_ depletion showed randomly extended protrusions and displayed a round body with an extended uropod [76].

Phosphatidylinositol 4 kinase type III α (PI4KIIIα) is an enzyme that produces phosphatidylinositol 4-phosphate (PI4P), which is important for plasma membrane identity, cell survival, and cell polarity [79]. TTC7A is an interacting partner of PI4KIIIα. Deficiency of TTC7A in lymphocytes led to the downregulation of PI3K/Akt/RhoA signaling and an imbalance in actin cytoskeleton dynamics [79]. The TTC7A-deficient cells demonstrated increased migratory speed compared with TTC7A-sufficient cells. These cells showed increased F-actin polymerization at the rear end, confirming that TTC7A was a significant regulator of actin polymerization in lymphocytes. The deficiency of TTC7A led to reduced kinase activity of PI4KIIIα, thus reducing the levels of PI4P, which promoted actin filament polymerization and enhanced lymphocyte motility [79]. PI4P was also reported to play an important role in leukocyte cytoskeletal polarization. A local increase in plasma membrane curvature can trigger the recruitment of SRGAP2 (SLIT-ROBO Rho GTPase Activating Protein 2), which activates phosphatidylinositol 4-kinase alpha (PI4KA) to produce PI4P [80]. The polarized PI4P accumulation is targeted by RPH3A (Rabphilin 3A), which recruits PIPKIγi2 and phosphorylated myosin light chain. These events control neutrophil polarization and their adhesion to endothelium [80].

PIKfyve is another phosphoinositide kinase that generates PI3,5P_2_ and PI5P [81]. Treatment with the PIKfyve inhibitor apilimod affected neutrophil chemotaxis and migration toward the fMLP gradient. Higher concentrations of apilimod limited the distance that the neutrophils traveled and caused a severe reduction in neutrophil migratory speed. PIKfyve may collaborate with phosphoinositide 3-phosphatase MTMR3 (Myotubularin-Related Protein 3) to generate PI5P, which in turn stimulates Rac, thus helping in neutrophil migration [81,82].

## 5. Conclusions and Future Directions

This review discusses the significance of phosphoinositide production and conversion in the migration of immune cells. As elaborated in the previous sections, multiple phosphoinositide signaling pathways can play an important role in regulating immune cell migration. However, only the functions of PI3,4,5P_3_ and class I PI3Ks have been extensively investigated. To date, the function of other phosphoinositide signaling has received less attention. Further in-depth research is required to fully answer the following questions: 1. Which phosphoinositide kinases and phosphatases are responsible for the control of phosphoinositide levels during immune cell migration? 2. Which phosphoinositide effectors are recruited to mediate a specific phosphoinositide function? 3. What is the impact of phosphoinositide signaling in a specific immune response or immune disorder?

The effects of phosphoinositide signaling on immune cell migration could be context-dependent. One reason is the complexity of immune cells. Different types of immune cells, their maturity status, and challenges by different pathogens or cytokines may lead to different dependence on phosphoinositide signaling during cell migration. Another reason is that immune cells may respond differently under different experimental conditions such as in vitro vs. in vivo, 2D vs. 3D, migration on different surfaces, or migration induced by different chemoattractants. Therefore, investigating a specific type of immune cell under specific pathological or experimental conditions should be carried out with caution as our understanding of phosphoinositide signaling function is limited. Validation of the results from in vitro experiments in in vivo models is encouraged to confirm the pathological impact.

Further research of phosphoinositide signaling in modulating immune cell migration and regulating immune cell function may provide clues as to whether the components of phosphoinositide signaling have the potential to be targeted to treat immune disorders. The infiltration of dendritic cells and cytotoxic CD8^+^ T cells into tumors can mediate an anticancer immune response [83,84]. On the contrary, the recruitment of myeloid-derived suppressor cells (MDSCs) in the tumor microenvironment is immunosuppressive [85]. However, it remains unclear whether different phosphoinositide signaling can control immune cell tumor infiltration. As a future direction, research to explore the function of phosphoinositide signaling in immune cell recruitment to the tumor microenvironment and the effects on the cancer-immune response is warranted.

## Figures and Tables

**Figure 1 biomolecules-13-01705-f001:**
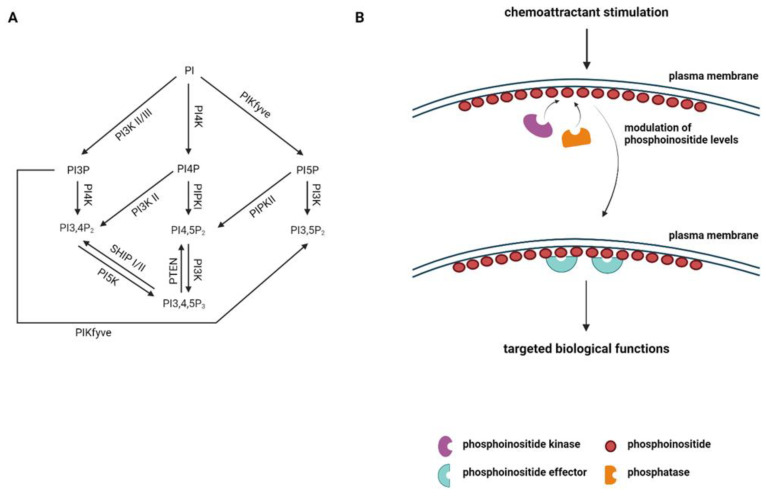
Phosphoinositide signaling pathway. (**A**) Schematic representation of the phosphoinositide (PI) family. The enzymes participating in the conversion of one form of PI to another are shown. (**B**) Schematic representation of regulation of phosphoinositides on chemoattractant stimulation is depicted. Phosphoinositide kinases and phosphatases control the specific phosphoinositide levels in the presence of a chemoattractant, which regulates the recruitment of phosphoinositide effectors to carry out targeted biological functions.

**Figure 2 biomolecules-13-01705-f002:**
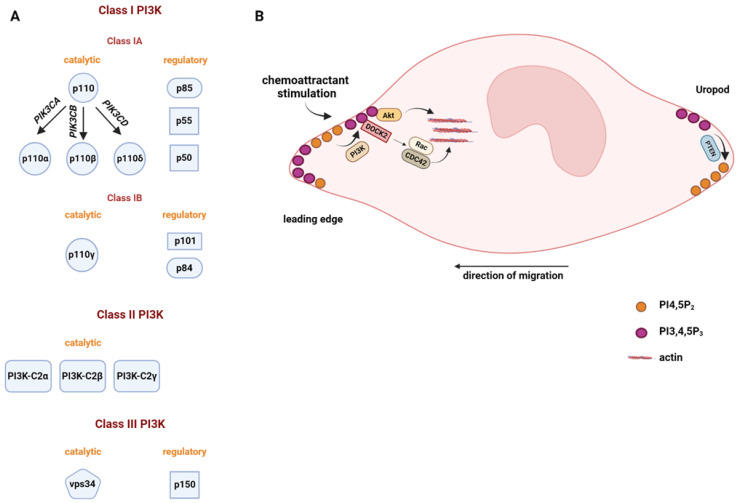
Function of PI3K family kinases in immune cell migration. (**A**) Different classes of the PI3K family kinases. Specific catalytic and regulatory subunits of different classes of PI3K are shown. (**B**) Chemoattractants stimulate PI3K activation at the leading edge, where the production of PI3,4,5P_3_ recruits its effectors such as Akt, DOCK2, Rac, and CDC42 to modulate cell polarization, protrusion formation, and cytoskeletal rearrangement. At the cell rear, PTEN hydrolyses PI3,4,5P_3_ to PI4,5P_2_ to maintain the PI3,4,5P_3_ front-to-rear gradient.

**Figure 3 biomolecules-13-01705-f003:**
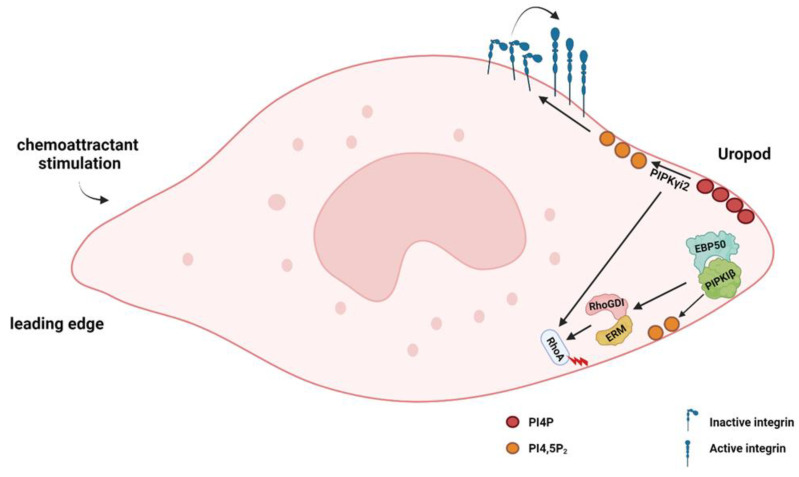
Function of PIPKIγi2 and PIPKIβ at the uropods in neutrophil migration. PIPKIγi2 and PIPKIβ are localized at the uropods during neutrophil migration. By producing PI4,5P_2_, PIPKIγi2 is required for neutrophil adhesion, integrin activation, and RhoA activation. By producing PI4,5P_2_ and interacting with EBP50, PIPKIβ enables ERM–RhoGDI interaction to trigger RhoA activation.

**Figure 4 biomolecules-13-01705-f004:**
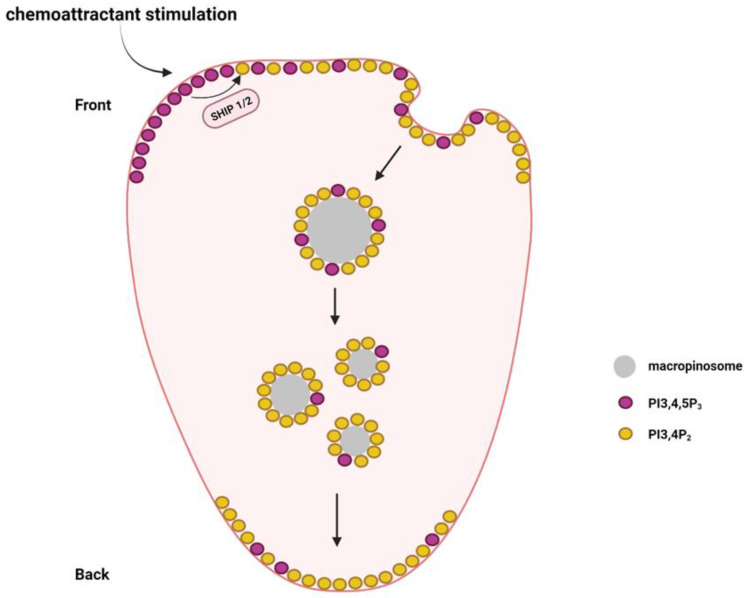
The establishment of a PI3,4P_2_ back-to-front gradient during neutrophil migration. SHIP1 and SHIP2 are required for the production of PI3,4P_2_ via dephosphorylation of PI3,4,5P_3_. The PI3,4P_2_ in the leading edge membrane can be transported to the rear by the formation of macropinosomes, which are sorted from the front-to-rear end.

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
