# Peer review of "Phosphoinositide Signaling in Immune Cell Migration"

_biomolecules, 2023, doi:10.3390/biom13121705_

Round 1
Reviewer 1 Report
Comments and Suggestions for Authors
This is a very well-written and structured review article. It is concise but gives all key outlines of knowledge to understand phosphoinositide signaling in the migration of leukocytes include neutrophil, mononuclear cells, T cells and B cells. I enjoyed reading this manuscript and learned a lot from it. I am only a little bit confused by Figure 1B in which phosphoinositide and its effectors look like protruded outside of the plasma membrane. Also, chemoattractant seems not interact with anything or directly located inside of the cells. Please modify this figure to make it easier to understand.
Author Response
Response to reviewers
We thank the Editor-in-Chief and reviewers for taking time out and going through our manuscript. We are extremely grateful for the feedback received and have incorporated the suggestions and critique. We are confident that the revised manuscript has addressed all the queries. Given below is the pointwise response to the reviewers’ comments.
Reviewer 1
This is a very well-written and structured review article. It is concise but gives all key outlines of knowledge to understand phosphoinositide signaling in the migration of leukocytes include neutrophil, mononuclear cells, T cells and B cells. I enjoyed reading this manuscript and learned a lot from it. I am only a little bit confused by Figure 1B in which phosphoinositide and its effectors look like protruded outside of the plasma membrane. Also, chemoattractant seems not interact with anything or directly located inside of the cells. Please modify this figure to make it easier to understand.
We would like to thank the reviewer for the positive feedback on the manuscript.
Fig 1B has been revised with the necessary changes and we hope that the revised figure clearly demonstrates that the chemoattractant is present extracellularly.
Reviewer 2 Report
Comments and Suggestions for Authors
The authors review the importance of phosphoinositides as signal mediators in adhesion, migration and polarization of various immune cells. Major attention goes to PI3-kinases. The review is well written with a clear focus on conversion of phosphoinositides and their biological effects in immune cells. An important aspect of phosphoinositide signaling is missing (see below). I also have major remarks on the figures.
Major remarks
11) An important aspect of phophoinositide signaling is not discussed in the current review: the hydrolysis of the head group. This is only briefly mentioned at the bottom of page 6 in one sentence. The authors should mention this important contribution of phophoinostide signaling in the introduction and explicitly state they will not discuss this. They can refer to the recent review on this type of signaling in immune cells by O’Donnell et al. Phospholipid signaling in innate immune cells J Clin Invest. 2018; 128(7): 2670-2679. https://doi.org/10.1172/JCI97944.
Perhaps also change the sentence in the abstract accordingly: “we discuss the function of different phosphoinositide signaling pathways in the migration of different types of immune cells”. It is only the conversion between different phosphoinositide forms that is discussed and not their hydrolysis to secondary messengers in signaling pathways. Also adapt the first sentence of the discussion, only conversion of phosphoinositide signaling is reviewed
22) Mandatory, improve the figures. Figures and schemes are important in reviews.
Figure 1B is misleading. First it gives the impression that the phospholipids are in membranes of organels (subcellular compartments) whereas most of their actions take place at the plasma membrane. In the cartoon the signal is even arriving at(in?) the subcellular compartment. Second the lipid tails are inserted in the membrane and are not on top of membranes as drawn. Third the action of the kinases and phosphatases do not result in release of the phosphoinositide from the membrane to find a target in the cytosol. Targets are recruited to the membranes. Similarly in Figure 4, is the conversion from PI(3,4,5)P3 to PI(3,5)P2 really occurring in the cystosol as suggested? Idem Figure 2 PTEN action.
Use the same color scheme and symbols including notation throughout the figures: chemoattractant are blue (figure 2 and 3) and purple circles (figure 4). PI(4,5)P2 - indicated as PIP2 - red circle (Figure 2) PI(4,5)P2 - indicated as PI(4,5)P2 - blue circle (Figure 3). PI(3,4,5)P3 - indicated as PIP3 – blue pentagon (Figure 3). PI(3,4,5)P3 - indicated as PI(3,4,5)P3 – violet circles (Figure 4).
Author Response
Response to reviewers
We thank the Editor-in-Chief and reviewers for taking time out and going through our manuscript. We are extremely grateful for the feedback received and have incorporated the suggestions and critique. We are confident that the revised manuscript has addressed all the queries. Given below is the pointwise response to the reviewers’ comments.
The authors review the importance of phosphoinositides as signal mediators in adhesion, migration and polarization of various immune cells. Major attention goes to PI3-kinases. The review is well written with a clear focus on conversion of phosphoinositides and their biological effects in immune cells.
We would like to thank the reviewer for taking time out to review our manuscript and for the positive feedback on the writing.
Major remarks
- An important aspect of phophoinositide signaling is not discussed in the current review: the hydrolysis of the head group. This is only briefly mentioned at the bottom of page 6 in one sentence. The authors should mention this important contribution of phophoinostide signaling in the introduction and explicitly state they will not discuss this. They can refer to the recent review on this type of signaling in immune cells by O’Donnell et al. Phospholipid signaling in innate immune cells J Clin Invest. 2018; 128(7): 2670-2679. Perhaps also change the sentence in the abstract accordingly: “we discuss the function of different phosphoinositide signaling pathways in the migration of different types of immune cells”. It is only the conversion between different phosphoinositide forms that is discussed and not their hydrolysis to secondary messengers in signaling pathways. Also adapt the first sentence of the discussion, only conversion of phosphoinositide signaling is reviewed.
We thank the reviewer for the comments. The hydrolysis of the head group by phospholipases and the production of secondary messengers are also important aspects of phosphoinositide signaling pathway in modulating immune cell migration. The significance of phospholipases and production of secondary messengers in the immune system has been well defined in other reviews by O’Donnell et al. and Schilke et al (Reference list page 11, 429–432). In the revised manuscript, we have briefly discussed these reviews (page 4, lines 62–71) and stated that we will not discuss this part further in the current review. We have also made the necessary changes in both the Abstract (page 1, lines 16–17) and Conclusions (page 17, line 367–368) sections.
2) Figure 1B is misleading. First it gives the impression that the phospholipids are in membranes of organels (subcellular compartments) whereas most of their actions take place at the plasma membrane. In the cartoon the signal is even arriving at(in?) the subcellular compartment. Second the lipid tails are inserted in the membrane and are not on top of membranes as drawn. Third the action of the kinases and phosphatases do not result in release of the phosphoinositide from the membrane to find a target in the cytosol. Targets are recruited to the membranes. Similarly in Figure 4, is the conversion from PI(3,4,5)P3 to PI(3,5)P2 really occurring in the cystosol as suggested? Idem Figure 2 PTEN action.
We thank the reviewer for bringing this to our notice. We have corrected Figure 1B. We apologize for the misrepresentation in Figure 4. The conversion from PI3,4,5P3 to PI3,4P2 occurs in the plasma membrane and not in the cytosol. PI3,4P2 also localizes to nascent macropinosomes that are internalized from leading edge membrane. The macropinosomes further break up into smaller PI3,4P2-enriched vesicles, which then fuse with the plasma membrane at the rear of the cell. We have updated Figure 4 to put across our point correctly in the revised figure.
Use the same color scheme and symbols including notation throughout the figures: chemoattractant are blue (figure 2 and 3) and purple circles (figure 4). PI(4,5)P2 - indicated as PIP2 - red circle (Figure 2) PI(4,5)P2 - indicated as PI(4,5)P2 - blue circle (Figure 3). PI(3,4,5)P3 - indicated as PIP3 – blue pentagon (Figure 3). PI(3,4,5)P3 - indicated as PI(3,4,5)P3 – violet circles (Figure 4).
We thank the reviewer for the suggestion. We have incorporated the same color scheme and shapes for the elements across all the figures to maintain consistency.
Round 2
Reviewer 2 Report
Comments and Suggestions for Authors
Adequately revised, clear improvement of figures.